# Predictors of neonatal sepsis in public referral hospitals, Northwest Ethiopia: A case control study

**Tadesse Yirga Akalu**[1]*, **Bereket Gebremichael**[2], **Kalkidan Wondwossen Desta**[2], **Yared Asmare Aynalem**[3], **Wondimeneh Shibabaw Shiferaw**[3], **Yoseph Merkeb Alamneh**[4]

**1** College of Health Science Debre Markos University, Debre Markos, Ethiopia, **2** College of Health Science, Addis Ababa University, Addis Ababa, Ethiopia, **3** College of Health Science, Debre Berihan University, Debre Berihan, Ethiopia, **4** Scool of medicine, Debre Markos University, Debre Markos, Ethiopia

* tadesseyirga680@gmail.com

**Data Availability Statement:** All relevant data are within the manuscript and its Supporting Information files.

## Abstract

### Background

Despite remarkable progress in the reduction of death in under-five children, neonatal mortality has shown little or no concomitant reduction globally. It is also one of the most common causes of neonatal death in Ethiopia. Little is known on predictors of neonatal sepsis. Risk based screening and commencement of treatment appreciably reduces neonatal death and illness. Therefore, the main aim of this study was to identify predictors of neonatal sepsis in public referral hospitals of Northwest Ethiopia.

### Methods

Institutional based unmatched case-control study was conducted among a total of 231 neonates in Debre Markos and Felege Hiwot referral hospitals from March 2018- April 2018. Neonates who fulfill the preseted criteria for sepsis were considered as cases and neonates diagnosed with other medical reasons except sepsis were controls. For each case, two consecutive controls were selected by simple random sampling method. Data were collected using structured pretested questionnaire through a face to face interview with index mothers and by reviewing neonatal record using checklist. The collected data were entered into Epi data version 3.1 and exported to STATA/ SE software version 14. Binary and multivariable logistic regression analyses were employed. Statistical significance was declared at P<0.05.

### Result

Multivariable logistic regression analysis showed that, duration of rupture of membrane $\geq$ 18hours was significantly associated with sepsis (AOR = 10.4, 95%CI = 2.3–46.5). The other independent predictors of neonatal sepsis were number of maternal antenatal care service $\leq$3 (AOR = 4.4, 95%CI = 1.7–11.5), meconium stained amniotic fluid (AOR = 3.9, 95%CI = 1.5–9.8), urinary tract infection during pregnancy (AOR = 10.8, 95% CI = 3.4–33.9), intranatal fever (AOR = 3.2, 95% CI = 1.1–9.5), first minute APGAR score <7 (AOR = 3.2, 95% CI = 1.3–7.7), resuscitation at birth (AOR = 5.4, 95% CI = 1.9–15.5), nasogastric tube insertion (AOR = 3.7, 95% CI = 1.4–10.2).

**Funding:** The author(s) received no specific funding for this work.

**Competing interests:** The authors have declared that no competing interests exist.

**Abbreviations:** ANC, Antenatal Care; APGAR, Activity, Pulse, Grimace, Appearance, Respiration; CBC, Complete Blood Count; CI, Confidence Interval; EONS, Early Onset Neonatal Sepsis; LONS, Late Onset Neonatal Sepsis; MSAF, Meconium Stained Amniotic Fluid; NICU, Neonatal Intensive Care Unit; OR, Odds Ratio; PIH, Pregnancy Induced Hypertension; PROM, Prolonged Rupture Of Membrane; RR, Relative Risk; SVD, Spontaneous Vaginal Delivery; UTI, Urinary Tract Infection.

## Conclusion

Neonatal invasive procedures, ANC follow up during pregnancy, different conditions during birth like meconium stained amniotic fluid, low APGAR score and resuscitation at birth were the independent predictors of neonatal sepsis.

## Background

Neonatal sepsis is a common critical illness in neonate. It is outlined as a clinical condition characterized by a syndrome of infection by the presence of clinically suspected or culture confirmed infection in the first 28 days after birth [1–3]. It comprises abundant systemic infections of the neonate like septicemia, meningitis, pneumonia, arthritis, urinary tract infection but it does not embrace muco-cutaneous infections like conjunctivitis and oral thrush [2].

As indicated in different literatures, neonatal sepsis is caused by both maternal and neonatal factors [4–6]. Delivery by caesarian section, male sex and prematurity were identified as risk factors of neonatal sepsis [6, 7]. Maternal factors such as intra-partum fever, instrumental delivery and place delivery are significant predictors of neonatal sepsis. Neonates born from women with more than three times digital per vaginal examination and never attend antenatal care (ANC) are at higher risk for neonatal sepsis [8–10].

Sepsis in neonates is among the leading causes of neonatal mortality and morbidity, especially in the first one week of life in low and middle-income countries (LMIC) [11, 12]. Globally about four million neonates die in a year, from this 98% are from developing countries particularly from sub-Saharan Africa[13]. The effect of neonatal death is assumed to be six times more in the developing countries compared to developed [14]. Around 6.9 million infants reported to have sepsis every year in sub-Saharan Africa, south Asia, and Latin America [11]. In 2015, from 5.941 million deaths in under-five years old children, forty-five percent of them died in the neonatal period. The death is greater than 50% in lots of regions, including Tanzania, Uganda, Congo and India [12].

In Africa, 17% among all neonatal death results from neonatal sepsis as compared to only six percent in developed countries. To enhance the existence of neonates, works should be targeted to reduce neonatal sepsis with a focus in sub-Saharan Africa and South Asia [15, 16].

Based on a recent community based study in rural Ethiopia, sepsis is the leading cause of death [17]. According to Ethiopian Demographic health survey (EDHS) 2016 report, the neonatal mortality rate (NMR) was 29/1000 live births, which has no significant reduction from the 2011 EDHS report which was 37/1000 live births. This significant number of death is greatly attributed to neonatal sepsis [18, 19]. To achieve sustainable development goal (SDG), reducing newborn and under five mortalities as low as 12/1000 and 25/1000 respectively, is one of the Global strategies of WHO in African countries by 2030. This could be achieved through better prevention and management of preterm births and severe infections [20]. Identification of risk factors and timely initiation of treatments, can significantly decrease neonatal mortality and morbidity [15].

Despite the presence of few studies regarding risk factors and etiology of neonatal sepsis, there are some contradicting or inconsistent findings on some predictors for neonatal sepsis, like neonatal sex, prematurity, low birth weight and residence. In addition some factors specifically neonatal invasive procedures were not incorporated [5, 6, 21]. Moreover, as far as literature search showed, there is no study conducted in the study area regarding predictors of

neonatal sepsis. Therefore, this study was intended to identify predictors of neonatal sepsis in NICU of public referral hospitals of Northern Ethiopia.

## Materials and methods

### Study area and period

The study was conducted in the referral Hospitals of East and West Gojjam zones, Amhara Region, Ethiopia, from March 2018- April 2018. Felege Hiwot Referral Hospital (FHRH) found in Bahir Dar, the capital of Amhara regional state, whereas Debre markos referral hospital (DMRH) found in Debre Markos, the town of East Gojjam Administrative Zone. DMRH and FHRH are found 299 Kilometers and 565 Kilometer far from Addis Ababa (capital City of Ethiopia) respectively. According to information obtained from the administrative offices of the hospitals, they provide different services in the outpatient department, inpatient department and operation room theatre department. DMRH and FHRH serve for more than 3.5 million and 5 million populations in their catchment area respectively. FHRH had six physicians and 20 nurses and 30 beds in NICU with total annual neonatal admission of more than 3500 of which more than 800 was due to sepsis. DMRH had 20 beds with annual admission of above1400 neonates of which, more than 400 neonates were diagnosed with neonatal sepsis. There were three physicians and 21 nurses in NICU.

### Study design and population

Institutional based unmatched case control study was conducted among cases (neonates with sepsis) and control (neonates without sepsis) admitted in Neonatal Intensive Care Units (NICU) in referral hospitals of Northern Ethiopia. All neonates who were admitted to Neonatal Intensive Care Units were our study population.

**Cases.** were considered with the presence of one of the seven clinical signs and two or more hematologic criteria suggestive for neonatal sepsis. Neonates with clinical signs of possible serious bacterial infection (PSBI), according to the Young Infants Clinical Signs Clinical Study(YICSS) criteria of WHO's Integrated Management of Neonatal and Childhood Illness (IMNCI) guidelines, are defined as the presence of any one of a history of [difficulty feeding, history of convulsions, movement only when stimulated, respiratory rate of 60 or more breaths per min, severe chest retractions, or a temperature of 37.5˚C or higher or 35.5˚C or lower and change in level of activity][22]. Cyanosis and grunting have been included by another study [23]. Presence of any one of seven clinical signs and symptoms predict severe infection (based on an expert pediatrician's assessment) and was associated with a sensitivity and specificity of 85% and 75% respectively in 0–6 day old neonates and 74 and 79% respectively in infants aged 7–59 days[22].including others like bradycardia, tachycardia, irritability, oxygen requirement, increased frequency of apnea, poor capillary refill, along with ≥2 of the hematological criteria; total leukocyte count (<5000 or>12000 cells/ μl, absolute neutrophil count (<1500 cells/ μl or>7500 Cells/μl), erythrocyte sedimentation rate (ESR) (>15/1 h) and platelet count (<150 x10$^3$ or>450 x10$^3$ cells/ μl), elevated CRP > 1 mg/dl(>15mg/L), Glucose intolerance confirmed at least 2 times: hyperglycemia (blood glucose >180 mg/dL) OR hypoglycemia (glycaemia < 45 mg/dl) when receiving age specific normal range glucose amounts [22]. And who were admitted to NICUs.

**Controls.** neonates who were not diagnosed as neonatal sepsis (do not fulfill sepsis criteria) and admitted with other medical reasons in the two referral hospitals. Cases and controls were identified through history taking and physical examination using prepared clinical sign checklists for the neonates during the study period in the NICU. The diagnosis includes history taking, clinical manifestations (objective findings), and laboratory tests.

## Sample size determination and sampling procedure

The sample size was determined using Epi-Info version7. Double population proportion exposure difference formula was used by using place of delivery (at health center comparing to those who gave birth in hospitals) as independent predictor exposure variable, since it gave the maximum sample size. From that study, the proportion of women among controls with health center delivery was 8.3% and the proportion of women among cases with health center delivery was 26.9% [5]. 1 to 2 cases to controls ratio was recruited to achieve 90% power at 5% significance level. Adding 10% non-response rate, the total sample size was 231 with 77 cases and 154 controls. Cases were selected consecutively among those neonates admitted in the neonatal intensive care units (NICUs). The next immediate two corresponding controls were selected by simple random method at the same day and in the same neonatal intensive care unit.

## Data collection tool and procedure

Data collection tool was first developed in English and were translated in to local language (Amharic) and was translated back to English. Review was made by Amharic, English language experts and health professionals for consistency of language translation. A questionnaire was prepared by reviewing different literatures and other checklists which were related to risk factors of neonatal sepsis. Most questions were adopted from questionnaires used in other studies to investigate risk factors for neonatal sepsis [4, 5, 17]. Data were collected through record review (laboratory results: -like CBC, CRP, ESR,) and a face to face interview of the index mothers using pretested structured and interviewer administered questionnaires by trained experienced health professionals. They were interviewed about their socio-demographic characteristics, maternal factors and neonatal factors by trained health personnel in each NICU in the two public referral hospitals. Data were collected by four trained BSc nurses and they were supervised by the investigators. Continuous follow up and supervision was made by the principal investigator throughout the data collection period.

## Data quality management

Two days training was given for data collectors and supervisors about the objective of the study, data collection tool, data collection procedures, ethical consideration during data collection. A pretest was conducted on 5% of the sample size before actual data collection and necessary adjustment was made on the tool. Pretest was done in Finote Selam district hospital, which is found in West Gojjam. Close supervision was carried out by the investigators during the data collection time. Data from each respondent were checked for its completeness, clarity, consistency and accuracy by the supervisor.

## Data processing and analysis

Data were checked for completeness and consistencies and then it was cleaned, coded and entered using Epi data version 3.1 and it was exported to STATA version 14 for analysis. Descriptive statistics were used to describe the study population in relation to relevant variables. Chi-square and odds ratio (OR) were used to assess the relationship between factors associated with the occurrence of neonatal sepsis. Then variables that had association in the bivariate model (p<0.25) were entered and analyzed by a multivariable logistic regression model to identify the independent effect of different factors for occurrence of sepsis. Statistical significance was declared at P<0.05.

## Ethics approval and consent to participate

Ethical clearance was obtained from Addis Ababa University, college of Health science ethical review board. Then officials at different levels in the hospitals were communicated through formal letters. The responsible bodies at neonatal intensive care unit were told about the purpose of the study and written informed consent was obtained from participants to confirm willingness. They were notified that they have the right to refuse or terminate at any point of the interview. Confidentiality of the information was secured throughout the study process.

# Results

## Socio-demographic characteristics of the respondents

This study was intended to assess predictors of neonatal sepsis in public health facilities of East and West Gojjam, Ethiopia. A total of 231 neonates (77 cases and 154 controls) who were admitted in NICU with their mothers were included with the overall response rate of 100%. According to this study, the mean age of neonates was 8(SD ±6.1) days. The mean age of mothers was 28.2 (SD ±6.4) years. Most of the participants were from urban areas (63.6% cases and 61.0% controls) and more than half 45(58.4% cases were males and 74(51.9%) of controls were females. Most of the study participants were Orthodox Christian followers. Concerning marital status of mothers, 66(85.7%) of cases and 133(86.4%) controls were married (Table 1).

## Descriptive statistics of maternal factors for neonatal sepsis

This study revealed that, most of mothers 57(74.0%) of cases and 133(86.4%) of controls had ever got ANC service whereas 20(26.0%) of cases and 21(13.6%) of controls had never got ANC service during their pregnancy of the current neonate. The proportion of women who got ANC service less than three times is higher in cases 23(40.4%) than controls 25(18.8%)

**Table 1. Socio demographic characteristics of neonates and their mothers.**

| Exposure Variables | Responses/category | Cases | Controls |
|---|---|---|---|
| | | Count (%) | Count (%) |
| Marital status | Single | 6(7.8) | 14(9.1) |
| | Widow | 5(6.5) | 7(4.5) |
| | Married | 66(85.7) | 133(86.4) |
| Maternal religion | Orthodox | 59(76.6) | 103(66.9) |
| | Muslim | 13(16.9) | 42(27.3) |
| | Protestant | 5(6.5) | 9(5.8) |
| Maternal residence | Urban | 49(63.6) | 94(61.0) |
| | Rural | 28(36.4) | 60(39.0) |
| Maternal educational status | unable to read and write | 24(31.2) | 52(33.8) |
| | Primary | 27(35.1) | 39(25.3) |
| | Secondary | 13(16.9) | 36(23.4) |
| | college and higher | 13(16.9) | 27(17.5) |
| Neonatal sex | Male | 45(58.4) | 74(48.1) |
| | Female | 32(41.6) | 80(51.9) |
| Maternal age | ≤20 | 15(19.5) | 13(8.4) |
| | 21–34 | 46(60.0) | 110(71.4) |
| | ≥35 | 16(20.8) | 31(20.1) |
| Neonatal age | <7 | 33(43.9) | 98(63.6) |
| | 7–28 | 44(57.1) | 56(36.4) |

similarly the proportion of women with duration of labor after rupture of membrane >18hrs is higher in cases 20(26.0%) than controls 3(1.9%). More than half of women had given birth at hospital 4(57.1%) of cases and 83(53.9%) of controls. The proportion of women who had intrapartum fever was three times higher in cases 33(42.9%) than controls 22(14.3%). Regarding mode of delivery, more than half 102(66.2%) controls and 41(53.2%) of cases had spontaneous vaginal delivery. Women who had meconium stained amniotic fluid, were more 43 (55.8%) in the cases than controls 35(22. 7%).The proportion of women who had pregnancy induced hypertension was higher in cases 15(19.5%) compared to controls 16(10. 4%).Similarly, women with a history of urinary tract infection (UTI) during their pregnancy was eight times higher among cases 29(37.7%) than controls 7(4.5%).

## Descriptive statistics of neonatal factors for neonatal sepsis

In this study more than half of 42(54.5%) cases and nearly three fourth of controls 110(71.4%) were in the gestational age of 37–42 completed weeks, whereas the proportion of neonates with gestational age <32 completed weeks were higher in cases 8(10.4%) than controls 6 (3.9%). Similarly, the proportion of neonates with first and fifth minute APGAR score <7 was 45(60.0%) and 24 (32.0%) in cases respectively which was higher than controls 23(15.9%), 6 (4.1%) in first and fifth minute respectively. Similarly, the proportion of neonates with birth weight 1500-2500grams was higher in cases 35(45.5%) than controls 43 (27.9%). More than half of cases 46 (59.7%) and less than one fourth of controls 25(16.2%) have been resuscitated at birth.

## Factors associated with neonatal sepsis

The multivariable logistic regression result showed that, number of maternal ANC service was found to be significantly associated with neonatal sepsis. Those women who had ANC service ≤3 times were about four times more likely to have neonates suffered from sepsis compared to women who got ANC service >3 times (AOR = 4.4, 95%CI = 1.7–11.5). Similarly, duration of rupture of membrane was significantly associated with neonatal sepsis. Women with duration of rupture of membrane ≥ 18hours were around ten times more likely to have neonates with sepsis compared to women with duration of rupture of membrane <12 hours (AOR = 10.4, 95%CI = 2.3–46.5).

Meconium Stained Amniotic Fluid (MSAF) was found to be significantly associated with the risk of neonatal sepsis. Particularly, neonates delivered from women with meconium stained amniotic fluid were nearly four times more likely to develop sepsis compared with those neonates delivered from women without MSAF (AOR = 3.9, 95% CI = 1.5–9.8). Similarly, maternal urinary tract infection (UTI) and maternal fever during labor were significantly associated with the risk of neonatal sepsis. Particularly women with UTI were nearly eleven times more likely to have neonates suffering from neonatal sepsis compared to those without UTI (AOR = 10.8, 95% CI = 3.4–33.9). In addition to this, those women with fever during labor were around three times more likely to give birth to neonates who suffered from neonatal sepsis compared to those without intranatal fever (AOR = 3.2, 95% CI = 1.1–9.5).

Neonatal variables, first minute APGAR score <7 was found to be significantly associated with the risk of neonatal sepsis. Neonates with the first minute APGAR score <7 were approximately three times more likely to suffer from neonatal sepsis compared to neonates whose first minute APGAR score was ≥7 (AOR = 3.2, 95% CI = 1.3–7.7). Resuscitation at birth was also significantly associated with the risk of neonatal sepsis. Those neonates who had been resuscitated at birth were around five times more likely to suffer from sepsis compared to those without resuscitation at birth (AOR = 5.4, 95% CI = 1.9–15.5). In addition, nasogastric tube

insertion was significantly associated with the risk of neonatal sepsis. Specifically, neonates who were with nasogastric tube (NGT) were around four times more likely to get sepsis compared to neonates without NGT (AOR = 3.7, 95% CI = 1.4–10.2) (Table 2).

## Discussion

This study assessed predictors for neonatal sepsis in public health facilities. Maternal variables include number of ANC services, duration of rupture of membrane ($\geq$18hours), Meconium stained amniotic fluid (MSAF), number of antenatal care ($\leq$3 times) and maternal fever during labor and neonatal variables like first minute APGAR score <7, nasogastric tube (NGT) insertion and resuscitation at birth were the independent predictors of neonatal sepsis.

Duration of rupture of membrane (>18hours) was significantly associated with the risk of neonatal sepsis. Specifically, neonates born from women with duration of labor after rupture of membrane >18hrs were approximately ten times more likely to suffer from sepsis compared with those neonates born from women with duration of rupture of membrane <12 hours. This finding is comparable with studies conducted in Mexico (2012), Pakistan (2014), Nepal (2006), Tigre; Ethiopia (2016), USA (2013), Israel (2006) and United Kingdom (2002)[5, 6, 24–28].This could be due to the fact that birth canal is colonized with aerobic and anaerobic pathogens that might cause ascending amniotic fluid infection and colonization of the neonate at birth. Mother to fetus transmission of bacterial agents that infect the amniotic fluid and birth canal might occur in uterus more commonly during labor and delivery, which results in neonatal sepsis (EONS) [29].

The number of maternal antenatal care service was found to be significantly associated with neonatal sepsis. Particularly neonates born from women who had antenatal care service $\leq$ 3 times during their pregnancy were around four times more likely to develop sepsis compared to neonates born from women who had ANC service >3 times during their pregnancy. This might be due to differences in understanding of maternal and other risk factors. Women who has full ANC service could have a better understanding and medical care of risk factors than those women with incomplete ANC service. Previous studies conducted in Uganda (2015), India (2016) also showed that women who ever had ANC service were less likely to have neonates suffering from sepsis [7, 8],

The other independent predictor of neonatal sepsis was meconium stained amniotic fluid (MSAF). Neonates delivered from women with meconium stained amniotic fluid were nearly four times more likely to develop sepsis compared with those without. This is consistent with studies conducted in Mexico (2012), Ghana(2014), South Africa (2012), India (2017), Nepal and Indonesia (2015) [6, 9, 10, 25, 30, 31]. This might be due to the fact that neonates delivered from women with meconium stained amniotic fluid are more liable to aspirate it and fill smaller air ways and alveoli in the lung. This in turn increases the multiplication of microbes that cause sepsis and predisposes to late onset neonatal sepsis (LONS) [32].

Similarly, maternal fever during labor had significant association with neonatal sepsis. Neonates born from women with intrapartum fever were nearly four times more likely to suffer with neonatal sepsis compared to those without fever. This is comparable with findings from Pakistan (2014), India (2017), United Kingdom (2002) and Tigre; Ethiopia (2016) [5, 10, 24, 28] which revealed that, fever during delivery was an independent predictor of neonatal sepsis. This might be explained by the fact that, women with fever is an indicator of local or systemic infections like Chorioamnionitis or urinary tract infection. This results in hematogenious spread and vertical transmission of pathogens to the newborn before or during labor and delivery which further results in neonatal sepsis. Similar to the studies conducted in Tigre (2016), Mexico (2012), India (2017), and Debrezeit Ethiopia (2014) [5, 6, 10, 33] maternal

**Table 2. Bivariate and multivariable logistic regression analysis result.**

| Exposure Variables | Responses | Cases | Controls | COR with 95%CI | AOR with 95%CI |
|---|---|---|---|---|---|
| | | Count (%) | Count (%) | | |
| Maternal parity | Nullipara | 26(33.8) | 37(24.0) | 2.1(1.1–4.3) | |
| | Para-one | 30(39.0) | 54(35.1) | 1.7(.9–3.2) | |
| | Multipara | 21(27.3) | 63(40.9) | 1 | |
| Neonatal age | <7 days | 33(43.9) | 98(63.6) | 1 | |
| | 7–28 days | 44(57.1) | 56(36.4) | 2.3(1.3–4.1) | |
| Maternal ANC follow up | Yes | 57(74.0) | 133(86.4) | 1 | |
| | No | 20(26.0) | 21(13.6) | 2.2(1.1–4.4) | |
| Number of ANC services | ≤3 | 23(40.4) | 25(18.8) | 2.9(1.5–5.8) | 4.4(1.7–11.5)** |
| | >3 | 34(59.6) | 108(81.2) | 1 | 1 |
| Mode of delivery | Cs | 20(26) | 10(6.5) | 4.5(2.1–11.5) | |
| | Instrumental | 16(20.8) | 42(27.3) | 1.1(.6–2.2) | |
| | SVD | 41(53.2) | 102(66.2) | 1 | |
| Duration of rupture of membrane | <12 hrs. | 25(32.5) | 121(78.6) | 1 | 1 |
| | 12-18hrs | 30(39.0) | 28(18.2) | 5.2(2.7–10.1) | |
| | >18hrs | 22(28.5) | 5(3.2) | 4.1(1.4–12.3) | 10.4(2.3–46.5)** |
| Meconium stained amniotic fluid | Yes | 43(55.8) | 35(22.7) | 4.3(2.4–7.7) | 3.9(1.5–9.8)** |
| | No | 34(44.2) | 119(77.3) | 1 | 1 |
| Number of PV examination | ≤3 | 21(27.3) | 88(57.5) | 1 | |
| | >3 | 56(72.7) | 65(42.5) | 3.6(2.0–6.6) | |
| Intra-natal fever | Yes | 33(42.9) | 22(14.3) | 4.5(2.4–8.5) | 3.2(1.1–9.5)** |
| | No | 44(57.1) | 132(85.7) | 1 | 1 |
| Foul smelling amniotic fluid | Yes | 19(24.7) | 7(4.5) | 6.9(2.8–17.2) | |
| | No | 58(75.3) | 147(95.5) | 1 | |
| History of UTI | Yes | 29(37.7) | 7(4.5) | 12.7(5.2–30.8) | 10.8(3.4–33.9)* |
| | No | 48(62.3) | 147(95.5) | 1 | 1 |
| First minute APGAR score | <7 | 45(60.0) | 23(15.8) | 8.0(4.2–15.2) | 3.2(1.3–7.7)** |
| | ≥7 | 30(40.0) | 123(84.2) | 1 | 1 |
| Fifth minute APGAR score | <7 | 24(32.0) | 6(4.1) | 10.9(4.3–28.4) | |
| | ≥7 | 51(68.0) | 140(95.9) | 1 | |
| Birth weight | ≥4000 gm. | 5(6.5) | 9(5.9) | 1.8(.6–5.8) | |
| | 2500–4000 gm. | 30(39.0) | 97(63.0) | 1 | |
| | 1500–2500 gm. | 35(45.4) | 43(27.9) | 2.6(1.4–4.8) | |
| | <1500 gm. | 7(9.1) | 5(3.2) | 4.5(1.3–15.3) | |
| Immediate cry | Yes | 37(48.1) | 113(73.4) | 1 | |
| | No | 40(51.9) | 41(26.6) | 2.9(1.7–5.3) | |
| Resuscitation at birth | Yes | 46(59.7) | 25(16.2) | 7.7(4.1–14.31) | 5.4(1.9–15.5)** |
| | No | 31(40.3) | 129(83.8) | 1 | 1 |
| Endotracheal intubation | Yes | 6 (7.8) | 5 (3.2) | 1 | |
| | No | 71 (92.2) | 149(96.8) | 2.5(0.7–8.5) | |
| Nasogastric tube insertion | Yes | 38(49.4) | 16(10.4) | 8.4(4.2–16.7) | 3.7(1.4–10.2)** |
| | No | 39(50.6) | 138(89.6) | 1 | 1 |
| Urinary catheter insertion | Yes | 8(10.4) | 5(3.2) | 3.5(1.1–10.9) | |
| | No | 69(89.6) | 149(96.8) | 1 | |

Key: CS Cesarean Section, SVD Spontaneous Vaginal Delivery, gm. Gram

* = p-value<0.001

** = p-value<0.05

urinary tract infection (UTI) was significantly associated with neonatal sepsis. Neonates born from women with UTI during their pregnancy were around eleven times more likely to suffer from sepsis.

According to this study APGAR score was one of the neonatal variable which was found to be significantly associated with neonatal sepsis. Neonates who had the first minute APGAR score < 7 were around three times more likely to be affected by neonatal sepsis compared to those neonates who had first minute APGAR score >7. This finding was similar with studies conducted in Ghana (2014), Indonesia (2015) and Nepal (2006) revealed that, first minute APGAR score less than seven was an independent risk factor of neonatal sepsis [25, 31, 34]. APGAR score is the overall indicator for the state of the newborn in the extra uterine environment and neonates. Low APGAR score, could be in a state of bradycardia, asphyxia and need emergency support. This might result in exposure for pathogenic microorganisms through unsterile assisting equipment.

In this study, resuscitation at birth was a significant risk factor for neonatal sepsis. Neonates who were resuscitated at birth were nearly five times more likely to develop sepsis compared to those who were not resuscitated. This finding is in line with findings from other studies: Ghana (2014), Mexico (2012) and Tanzania (2016) [6, 31, 35, 36]. These could be due to the fact that, if the procedure of resuscitation is done forcefully, it might cause laceration to the susceptible and easily breakable mucous membrane of the neonate and serve as a route of entry for pathogens from unsterile equipment [37]. In addition, nasogastric tube insertion (NGT) was found to be significantly associated with neonatal sepsis. Neonates who were on NGT feeding were around four times more likely to be affected by neonatal sepsis compared to those without NGT. This finding was similar with studies conducted in Turkey (2016) and Taiwan (2016). [38, 39]. This might be due to the possibility of easily accessibility of pathogenic organisms during the procedure.

## Strength and limitations of the study

We used case control study design which is appropriate to address the research question. The other strength of this study is it was conducted in relatively larger area. The cases and controls were also taken from the same institutions which made it easy to identify controls. This study also has certain limitations. Since the study participants were selected from institutions, neonates having signs and symptoms of sepsis who didn't come to hospitals for medical care might be missed resulting in reduced external validity. Besides, the identification of sepsis cases was not based on culture confirmed sepsis. However, it was based on suggestive clinical presentations and sepsis indicative laboratory findings.

## Conclusion

The findings of this study suggest that, among neonates admitted in neonatal intensive care units (NICU) of the two hospitals, both maternal and neonatal variables including some invasive procedures were significantly associated with neonatal sepsis. Maternal variables; (number of ANC services, duration of rupture of membrane (≥18hours), Meconium stained amniotic fluid (MSAF), number of Antenatal care (>3 times) and maternal fever during labor) and neonatal variables (first minute APGAR score <7, resuscitation at birth and nasogastric tube insertion) were significantly associated with neonatal sepsis. Therefore, Professionals working in NICUs are recommended to adhere to aseptic techniques while carrying out neonatal invasive procedures and enhance maternal education on risk factors like UTI, incomplete ANC service and should incorporate routine neonatal sepsis screening into the care of neonates and mothers. Neonates with low APGAR score and those born from women with intrapartum

fever should get special attention to prevent sepsis. Future researches on neonatal sepsis are recommender to include neonates in the community which may increase external a validity of the study.

## Supporting information

**S1 File.**
(DTA)

## Acknowledgments

First of all, our special thanks & deepest gratitude goes to Addis Ababa University for financial support to undertake this research work. We would also like to extend our heartfelt thanks for the study participants, data collectors and supervisor who participated in the study. We are also thankful for administrators of Hospitals, and head nurses of NICU in the respective Hospitals.

## Author Contributions

**Conceptualization:** Tadesse Yirga Akalu.

**Data curation:** Tadesse Yirga Akalu.

**Formal analysis:** Tadesse Yirga Akalu.

**Methodology:** Tadesse Yirga Akalu, Bereket Gebremichael, Kalkidan Wondwossen Desta, Yared Asmare Aynalem, Wondimeneh Shibabaw Shiferaw.

**Software:** Tadesse Yirga Akalu.

**Supervision:** Bereket Gebremichael, Kalkidan Wondwossen Desta.

**Validation:** Bereket Gebremichael, Kalkidan Wondwossen Desta, Yared Asmare Aynalem, Wondimeneh Shibabaw Shiferaw.

**Visualization:** Tadesse Yirga Akalu, Bereket Gebremichael, Yared Asmare Aynalem, Wondimeneh Shibabaw Shiferaw, Yoseph Merkeb Alamneh.

**Writing – original draft:** Tadesse Yirga Akalu, Bereket Gebremichael, Kalkidan Wondwossen Desta.

**Writing – review & editing:** Tadesse Yirga Akalu, Bereket Gebremichael, Kalkidan Wondwossen Desta, Yared Asmare Aynalem, Wondimeneh Shibabaw Shiferaw, Yoseph Merkeb Alamneh.

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
