## [Decision Letter · Decision Letter 0]

28 May 2020

PREDICTORS OF NEONATAL SEPSIS IN PUBLIC REFERRAL HOSPITALS, NORTH WEST ETHIOPIA: A CASE CONTROL STUDY

PONE-D-19-33191

Dear Dr. Tadesse Yirga

We are pleased to inform you that your manuscript has been judged scientifically suitable for publication and will be formally accepted for publication once it complies with all outstanding technical requirements.

With kind regards,

Anna Palatnik, M.D.

Academic Editor

PLOS ONE

2. Please provide additional details regarding participant consent. In the ethics statement in the Methods and online submission information, please ensure that you have specified (1) whether consent was informed and (2) what type you obtained (for instance, written or verbal). If your study included minors, state whether you obtained consent from parents or guardians. If the need for consent was waived by the ethics committee, please include this information.

4. Your ethics statement must appear in the Methods section of your manuscript. If your ethics statement is written in any section besides the Methods, please move it to the Methods section and delete it from any other section. Please also ensure that your ethics statement is included in your manuscript, as the ethics section of your online submission will not be published alongside your manuscript.

Reviewers' comments:

Reviewer's Responses to Questions

**Comments to the Author**

1. Is the manuscript technically sound, and do the data support the conclusions?

Reviewer #1: Yes

2. Has the statistical analysis been performed appropriately and rigorously? 

Reviewer #1: Yes

3. Have the authors made all data underlying the findings in their manuscript fully available?

Reviewer #1: Yes

4. Is the manuscript presented in an intelligible fashion and written in standard English?

Reviewer #1: Yes

5. Review Comments to the Author

Reviewer #1: The authors report on various maternal and neonatal factors associated with the risk of sepsis based on selected crtieria. Their findings are consistent with the literature on this topic, although not confirmed by culture, which is a limitation of the study -- but understandable in the setting of northwest Ethiopia at two referral hospitals. The information may be useful to regional practitioners, but does not add substantially to the literature. Nonetheless, the study is straightforward with a simple design and reasonable statistical analyses. As such, it is deserving of publication because of its relevance to similar settings globally. Perhaps a comment to this effect might signal to readers that, although the findings are generally confirmatory of known risk factors, and limited because of the lack of culture confirmation of sepsis to direct individualized management decisions, they may be helpful to apply in similar settings where resources are limited.

6. PLOS authors have the option to publish the peer review history of their article (what does this mean?). If published, this will include your full peer review and any attached files.

Reviewer #1: No

---

## [Editor Report · Acceptance letter]

3 Jun 2020

PONE-D-19-33191 

PREDICTORS OF NEONATAL SEPSIS IN PUBLIC REFERRAL HOSPITALS, NORTH WEST ETHIOPIA: A CASE CONTROL STUDY 

Dear Dr. Akalu:

I'm pleased to inform you that your manuscript has been deemed suitable for publication in PLOS ONE. Congratulations! Your manuscript is now with our production department. 

Kind regards, 

on behalf of

Dr. Anna Palatnik 

Academic Editor

PLOS ONE